# Effectiveness of internet delivered cognitive behaviour therapy provided as routine care for people in the depressed phase of bipolar disorder treated with Lithium

**Olav Nielssen** [1,2]*, **Lauren Staples**[2], **Eyal Karin**[3], **Rony Kayrouz**[2], **Blake Dear**[2,3], **Nickolai Titov**[2,3]

**1** Faculty of Medicine and Health Sciences, Macquarie University, Sydney, Australia, **2** MindSpot Clinic, Macquarie Health, Macquarie University, Sydney, Australia, **3** eCentreClinic, Macquarie University, Sydney, Australia

* olav.nielssen@mq.edu.au

**Data Availability Statement:** Full data cannot be shared as it contains proprietary information.

## Abstract

There is little research reporting the outcome of internet delivered cognitive behaviour therapy, (iCBT), which helps patients identify and modify unhelpful cognitions and behaviours, for the depressed phase of bipolar disorder as part of routine care. Demographic information, baseline scores and treatment outcomes were examined for patients of MindSpot Clinic, a national iCBT service who reported taking Lithium and their clinic records confirmed the diagnosis of bipolar disorder. Outcomes were completion rates, patient satisfaction and changes in measures of psychological distress, depression and anxiety measured by the Kessler-10 item (K-10), Patient Health Questionnaire 9 Item (PHQ-9), and Generalized Anxiety Disorder Scale 7 Item (GAD-7), compared to clinic benchmarks. Out of 21,745 people who completed a MindSpot assessment and enrolled in a MindSpot treatment course in a 7 year period, 83 reported taking Lithium and had a confirmed a diagnosis of bipolar disorder. Outcomes of reductions in symptoms were large on all measures (effect sizes > 1.0 on all measures, percentage change between 32.4% and 40%), and lesson completion and satisfaction with the course were also high. MindSpot treatments appear to be effective in treating anxiety and depression in people diagnosed with bipolar, and suggest that iCBT has the potential to overcome the under-use of evidence based psychological treatments of people with bipolar depression.

## Author summary

This study examines the outcome of a subset of patients in the depressed phase of bipolar disorder (BDd), who enrolled in treatment in a free transdiagnostic cognitive behaviour therapy (iCBT) based treatment course offered by a national digital mental health service (DMHS). The selection of patients based on self reported treatment with Lithium was not sensitive, but was highly specific. The results show that iCBT is just as effective for people

Deidentified data sufficient to confirm the main findings can be found in the supplementary file.

**Funding:** The authors received no specific funding for this work.

**Competing interests:** The authors have declared that no competing interests exist.

with BDd as other forms of depressive illness, and offers the tantalising prospect of a more efficient and effective way of treating BDd, which is the more disabling phase of that disorder for many patients. The results offer further support for the emerging evidence that antidepressant medication and psychotherapy operate on different neurological pathways.

## Introduction

By definition, bipolar disorder (BD) is diagnosed after an episode of mania or hypomania, although the average duration between onset of mood disorder and diagnosis is around six years [1], and people with BD typically spend three times as long in the depressed phase of the disorder as in the manic or hypomanic phases, resulting in significant morbidity and disability [2–4]. A recent nationwide register-based cohort study from Finland estimated that 7.4% of people treated for depression will be diagnosed with BD within 15 years [5]. There are a number of studies reporting differences in the clinical characteristics of unipolar depressive disorder and the depressed phase of BD [6–8], and it might even be possible to differentiate bipolar depression (BDd) from major depression using functional neuroimaging [9], although the diagnosis of both forms of depressive illness is ultimately made by the presence of the syndrome of symptoms of depression. However, once the diagnosis has been established, the depressed phase of BD is then assumed to be due to that condition. BDd is associated with a greater risk of suicide, and is assumed to be less amenable to psychological treatments than depression with other aetiologies [10–12], and treatment guidelines have tended to emphasise medication over psychological treatments [13–15]. However, there is a large body of evidence for adjunctive psychosocial treatments for BD [16,17], with the largest number of trials of treatments based on cognitive behaviour therapy (CBT)[18,19], and more recent treatment guidelines have recommended the addition of psychological treatments for BDd, as well as to engage people in the management of their condition and to protect against relapse [20].

Studies of CBT in BD have been hampered by the inclusion of patients in both phases of the disorder, and the focus on relapse rates and social and cognitive function as outcomes, rather than the measurement of the effect of treatment on symptoms of depression [21]. Although there are a number of individual trials reporting good results for CBT for BDd, the sample sizes have been comparatively small, and the findings of three recent meta-analyses are inconclusive. Chiang and associates included 1384 patients from 19 randomised controlled trials and found a small positive effect on depressive symptoms (g = -0.54, 95% CI -0.03 to -0.96) [18], whereas two other meta-analyses found improvements in treatment adherence and social function, and lower relapse rates, but no effect on symptoms of depression [16,22].

A large number of clinical trials have demonstrated the efficacy of treatments for anxiety and depression delivered via the internet by mental health professionals trained in the systematic delivery of this model of treatment, with results that are equivalent to high quality face to face care [23–25]. There have been a number of internet delivered treatment programs specifically for bipolar disorder [21,26–28] although they are mostly directed at education about the disorder, improving adherence to medication, promoting social recovery and preventing relapse, rather than specifically delivering CBT for the depressed phase of BD. An exception is the Mood Swings plus (MS plus) program, adapted from face to face CBT [29], which subsequent studies have confirmed to be effective in treating symptoms of depression in people with bipolar disorder [27]. However, to our knowledge no studies reporting the outcome for the depressed phase of BD of iCBT delivered as part of routine care rather than in samples of BD patients recruited for clinical trials.

The MindSpot Clinic (MindSpot) was established as part of the Australian Government's eMental Health strategy to improve the availability of mental health services for adults with anxiety and depression, particularly for people who experience barriers to traditional forms of mental health care. MindSpot (www.mindspot.org.au) provides free assessment, and offers seven treatment courses, including four transdiagnostic courses for anxiety and depression and three disorder specific courses. In its seven years of operation, MindSpot has provided services to more than 140,000 people, and more than 30,000 Australians have enrolled in one of the seven treatment courses.

We have reported the overall results of services provided at MindSpot [30–32], but have not yet described outcomes for subgroups such as those with anxiety and depression reporting the diagnosis of BD. Therefore, the aims of the present study were (1) to examine the demographic and symptom profiles patients who completed assessments at MindSpot and who were likely to have bipolar disorder, (2) to report on the treatment outcomes for people with probable BDd compared to clinic benchmarks.

## Method

### Study design and participants

This prospective uncontrolled observational cohort study examined data from people who enrolled in a MindSpot treatment course between 1st January 2013 and 31st December 2019. MindSpot does not target people with BD and consequently the screening assessment has not included questions about that diagnosis or about symptoms of the disorder. However, we know that people with BD have used MindSpot, including several who have swung to the manic phase of the disorder while completing the course, and the screening assessment does include questions about medication. For the purposes of this study the clinic records of all the patients who reported taking Lithium were then examined for the stated reason for taking Lithium and whether there was confirmation of the diagnosis of BD by a psychiatrist. Treatment with Lithium is not a sensitive method of detecting BD, because an increasing number of people with bipolar disorder are treated with other mood stabilisers and antipsychotic medication, or remain undiagnosed [12]. However, in Australia the prescription of Lithium is fairly specific for the diagnosis of BD, although Lithium is also sometimes prescribed as an adjuvant treatment for treatment resistant major depression and for mood instability without a clear diagnosis of BD. Hence, demographic information, completion rates, satisfaction and symptom scores at baseline and after treatment of the patients enrolled in a MindSpot course who reported taking Lithium (n = 124), and who then had entries in their medical records either confirming the diagnosis of BD or the reason for taking Lithium (n = 88, confirmed diagnosis of BD n = 83, prescribed Lithium for other reasons n = 5) were examined. Information confirming the reason for taking Lithium was not available for the remaining patients either because the clinicians did not ask or record the reason, or because the patients chose self directed treatment, with little or no clinician contact, which was available for part of the study period. The outcomes for patients taking Lithium and those with confirmed BD were then compared with those of all patients who commenced a treatment course in the first seven years of operation (N = 21,745).

The MindSpot assessment, the nature and delivery of the treatment courses, and the procedure for maintaining patient safety in remote treatment are described in detail elsewhere [30,31,33]. MindSpot delivers seven digital treatment courses, which were developed and validated in a series of randomized controlled trials at the Macquarie University online research clinic, the eCentreClinic (www.ecentreclinic.org). Four of these are based on transdiagnostic principles, recognising that people often simultaneously experience symptoms of anxiety and

depression, and that common psychological skills are used to treat these symptoms. They are Mood Mechanic (for ages 18–25 years), the Wellbeing Course (26–65 years), Wellbeing Plus (over 65 years of age), and the Indigenous Wellbeing Course (for Aboriginal and Torres Strait Islander people) [30,31,34–36]. These four interventions are evidence-based psychological treatment programs that are largely agnostic to the causes of depression, and include psycho-education about mediators and moderators of symptoms, cognitive therapy, behavioural activation, graded exposure, sleep training, communication and interpersonal skills, problem solving, and relapse prevention [37,38]. The assessment questionnaires and enrolment procedures were constantly improved, but the courses themselves did not change during the years of the study. MindSpot also offers disorder-specific courses for Obsessive Compulsive Disorder, Post Traumatic Stress Disorder, and chronic pain. Patients can choose a treatment course based on their presenting symptoms and age, and since the majority were adults seeking assessment and treatment for anxiety and depression, most elected to enrol in the Wellbeing Course.

All courses consist of five lessons delivered over eight weeks. Each lesson comprises a series of slides that presents the principles of psychological treatment for the target symptoms in text and images, using principles of instructional design comprising both didactic and case-based learning [37]. Courses are delivered online with weekly support from a therapist, either by phone, secure email (private messaging), or both. The therapist time required per patient per course was between 1.5 and 3 hours [39], which includes all contact with patients, reading and responding to patient messages, administration and therapist supervision during treatment and follow-up. Materials are available online, although up to 10% of patients elected to receive course materials in a printed workbook, sent by post. For part of the period in which this study was conducted, an entirely self-guided version of the Wellbeing Course was offered, the results of which will be reported separately elsewhere. Clinic services are provided at no cost to participants.

## Outcome measures

Symptom questionnaires were completed weekly, but symptoms at baseline and at completion were used to calculate effects because of the variable trajectory of response between patients and because it provided the best representation of treatment outcome. Symptoms were measured using the Kessler 10-Item Scale (K-10) as a measure of general psychological distress [39], the Patient Health Questionnaire 9-Item (PHQ-9) for depression [40] and the Generalized Anxiety Disorder Scale 7-Item (GAD-7) for symptoms of generalized anxiety [41]. The PHQ-9 closely follows the DSM-IV diagnostic criteria for depression, and a score of >10 indicates the presence of a disorder [40]. Course completion and response to questions about patient satisfaction were also reported.

## Statistical analysis

To account for missing data, estimated means obtained from Generalised estimating equation (GEE) models were used for post-treatment scores, for both the bipolar sample and the clinic benchmarks [32]. Treatment effect sizes from assessment to post-treatment were measured using Cohen's *d*, percentage change in symptom scores from assessment to post treatment, and an estimate of the number needed to treat (NNT) to achieve a 50% improvement in symptoms of depression are also reported. Deterioration rates were calculated based on an increase in the PHQ-9 and GAD-7 scores from baseline to post treatment of 6 and 5 respectively. Data were analysed using SPSS version 21.0. A significance level of .05 was used for all analyses.

### Ethical review

Ethical approval for the collection and use of the data was obtained from the Macquarie University Human Research Ethics Committee (5201200912) and registered on the Australian and New Zealand Clinical Trials Registry (ACTRN12613000407796). Patients are also provided with the Terms of Use explaining that non-identifiable, aggregated data could be used for reporting and service evaluation purposes.

## Results

### Bipolar patients at assessment

Between 1st January 2013 and 31st December 2019, a total of 96,018 patients completed an assessment at MindSpot and 21,745 commenced one of the treatment courses. Of these 124 reported taking Lithium, and 88 had entries in their clinic records confirming the reason for taking Lithium, of whom 83 (94.3%) had entries in their clinic records confirming the diagnosis of BD. Those with confirmed BD were older (43.8 years, SD 13.3 vs 39.8 years, SD 13.8) were slightly less likely to be female (66.3% vs 71.4%), and were less likely to be employed (49.4% versus 61.2%). They were more likely to be married or report holding a university degree. The proportion reporting suicidal thoughts and plans were higher than the clinic benchmarks (34.3% versus 24.9%, 3% versus 1.1%), although the number disclosing suicidal plans, 2 out of the 67 (3%) who answered that question was too small to analyse. (Table 1)

Of the 124 patients who reported taking Lithium, 83 of 88 (94.3%) had entries in the records confirming the diagnosis of BD had been made by a psychiatrist, including a proportion who reported admission to hospital for treatment of manic episodes. In a further 5 cases (5.7%) the records stated that Lithium had not been prescribed for bipolar disorder, and instead as an adjuvant treatment for depression or for emotional lability arising from other conditions, confirming that treatment with Lithium is fairly specific for bipolar disorder in Australia. In the remaining 36 cases there was no confirmation of the reason for the prescription of Lithium.

### Treatment outcomes

All of the patients taking Lithium were enrolled in a transdiagnostic course, mostly Wellbeing (97/124, 78.2%), but some in Wellbeing Plus for over 60 years (18, 14.5%), and Mood Mechanic for those aged 18 to 25 years (9, 7.3%), and none enrolled in the courses for PTSD, OCD or chronic pain. In the benchmark sample 7% were enrolled in PTSD, OCD and chronic pain courses, which also measured symptom scores on the PHQ-9, GAD-7 and K-10.

**Table 1. Demographic Information.**

|  | Benchmark [*] | Lithium treatment | Confirmed BD |
|---|---|---|---|
|  | N = 21,745 | N = 124 | N = 83 |
| Age (mean and SD) | 39.8 (13.8) | 44.6 (12.8) | 43.8 (13.3) |
| Proportion female | 71.4% | 66.9% (83/124) | 66.3% (55/83) |
| Employed | 61.2% | 47.9% (57/119) | 49.4% (40/81) |
| Married | 47.8% | 48.7% (58/119) | 46.3% (37/80) |
| University degree | 38.6% | 47.1% (56/119) | 48.8% (39/80) |
| Suicidal thoughts | 24.9% | 32.0% (32/100) | 34.3% (23/67) |
| Suicidal plan | 1.1% | 2.0% (2/100) | 3.0% (2/67) |

[*]Benchmark column shows results from all patients that started treatment between 2013 and 2019 and answered assessment questions (Titov et al,. 2020)

**Table 2. Treatment outcomes.**

| | Clinic Sample | Lithium Treatment | Confirmed Bipolar Diagnosis |
|---|---|---|---|
| **Completion and satisfaction:** | | | |
| Started treatment | N = 21,745 | N = 124 | N = 83 |
| Completed lessons (4 or more) | 66.6% | 66.1% (82/124) | 69.9% (58/83) |
| Would recommend to others | 96.6% | 95.9% (70/73) | 96.2% (51/53) |
| **Symptom scores at assessment** | | | |
| K-10 | 30.1 (6.9) | 31.9 (7.5) | 31.6 (7.3) |
| PHQ-9 | 13.6 (5.9) | 15.2 (6.4) | 15.0 (6.2) |
| GAD-7 | 12.0 (5.0) | 12.3 (5.3) | 12.5 (5.3) |
| **Symptom scores at post-treatment**[*] | | | |
| K-10 | 20.8 (6.2) | 24.5 (6.6) | 24.6 (6.8) |
| PHQ-9 | 6.5 (4.2) | 8.9 (4.7) | 9.1 (4.7) |
| GAD-7 | 5.7 (3.6) | 7.3 (3.9) | 7.5 (4.1) |
| **Effect sizes** | | | |
| K-10 | 1.4 (1.40–1.44) | 1.1 (.78–1.31) | 1.0 (.67–1.31) |
| PHQ-9 | 1.4 (1.37–1.41) | 1.1 (.85–1.39) | 1.1 (.74–1.39) |
| GAD-7 | 1.5 (1.42–1.47) | 1.1 (.81–1.34) | 1.1 (.74–1.37) |
| **Percentage changes** | | | |
| K-10 | 46·3% [45·9% - 46·7%] | 33.8% [27.8%– 39.8%] | 32.4% [25.1% - 39.7%] |
| PHQ-9 | 52·2% [51.6% - 52.8%] | 41.4% [34.0% - 48.9%] | 39.3% [30.5%– 48.2%] |
| GAD-7 | 52·5% [52.1% - 52·9%] | 40.0% [33.1% - 48.2%] | 40.0% [30.9% - 49.1%] |
| **Clinical deterioration** | | | |
| PHQ-9 | 1·4% (184/13058) | 0 | 0 |
| GAD-7 | 2·2% (282/13058) | 1.6% (2/124) | 1.2% (1/83) |

Post-treatment scores using Estimated Means from GEE models

[*]Benchmark column shows results from all patients that started treatment between 2013 and 2019 (Titov et al., 2020)

Symptom scores at assessment and post-treatment were slightly higher for the bipolar group. However, patients with BD who enrolled in treatment courses achieved good symptom reductions. Bipolar patients showed large effect sizes, of 1.0 on all symptoms (95% CI 0.67–1.39 for all measures), although the improvement in symptom scores was lower than the clinic benchmark of 1.4 to 1.5 (95% CI 1.37 to 1.47 for all measures)[32]. There were also large improvements as calculated by percentage change in the K-10 (32.4%, 95% CI 25.1% - 39.7%), PHQ-9 (39.4%, 95% CI 30.5% - 48.2%) and the GAD-7 (40.0%, 95% CI 30.9% - 49.1%), although these were also lower than the clinic benchmarks (Table 2). The reliable deterioration rates were 1.4% for the PHQ-9 and 2.2% for the GAD-7 for the whole sample, but nil for the PHQ-9 and 1.8% for GAD-7 in the BD group.

There was no difference in the amount of therapist contact, lesson completion rates were similar for bipolar and other patients (66.1% vs 66.6% respectively, no significant difference), and treatment satisfaction at post treatment as measured by responses to a question on whether the patient would recommend MindSpot to someone else was also very high (95.9% vs 96.6%, also not significantly different).

## Discussion

The main finding of this study is that people with clinically significant symptoms of depression, with a mean PHQ-9 score of 15, and who were prescribed Lithium and were probably in the depressed phase of BD, achieved improvements in symptoms of depression with iCBT delivered as part of routine care that were similar to the outcomes achieved by people with depression of other aetiologies. They also had similar rates of course completion and treatment satisfaction. The results add support to other studies showing iCBT is effective for treatment of the depressed phase of BD [27], and that iCBT delivered as part of routine care has the potential to treat depression in people with BD as effectively as depression with other causes.

The finding that iCBT delivered in an efficient and accessible way as part of routine care is effective in the depressed phase of BD is important, because people with BD spend three times as long in the depressed phase of the disorder, and medications used to treat the depressed phase of BD are often both ineffective, using the measure of the numbers needed to treat (NNT) to remission of between 4 and 7, and can also be harmful, using the number needed to harm (NNH) of between 3 and 9 [42,43]. The NNT for iCBT with a 50% reduction in symptoms is between 2 and 3, and the NNH based on reported deterioration rates, calculated as reliable change, is high, although the measurement of worsening of symptoms is not strictly comparable to the harm from side effects of medication. No antidepressant medication has received regulatory approval specifically for the treatment of BDd, and yet nearly half of all BD patients treated as outpatients are prescribed an antidepressant medication [44], despite the limited evidence for the efficacy of antidepressants in groups of patients with BDd [45]. Neuro-imaging studies suggest quite distinct neural changes in psychotherapy and treatment with antidepressant medication, with the changes associated with taking antidepressant (AD) medication being most apparent in the amygdala, at the centre of the limbic system that governs arousal, whereas the changes associated with psychotherapy are observed in the medial pre-frontal cortex, associated with cognitive processes and negative attribution [46]. Treatment outcomes for a sample of MindSpot patients taking antidepressant medication but were still depressed were just as good as for those who were not taking medication [47]. The results of this study suggest a prominent role for psychological treatments for BDd, and that wider use of iCBT could help to overcome the underuse of psychological treatments, and the distress and disability arising from BDd.

Users of MindSpot who were taking Lithium were less likely to be employed, consistent with the disabling effect of severe forms of mental illness, although they were more likely to report being married and having completed a university degree, possibly due to their older mean age. Although Lithium is still recommended as a first line treatment for BD, its use in Australia as a long term prophylactic treatment for BD has declined, as it has in the United States [44], and the older age of the sample screened on the basis of reported treatment with Lithium might also be due to different prescribing practices for more recently diagnosed BD patients. The results of this study suggest that treatment with Lithium is fairly specific for BD in Australia, as in the cases where the reason for its prescription was stated, 94% of patients taking Lithium understood they had been diagnosed with BD by a psychiatrist.

This study includes a number of significant limitations. The first is the information about the prescription of Lithium and other medication, and also that Lithium was in fact prescribed for confirmed BD, was self-reported, and there was no independent confirmation of the history of a manic episode. However, the sample size is quite large and most of the patients were contacted by telephone by MindSpot therapists in the course of assessment and treatment, many of the patients reported admissions to hospital for treatment of mania, and some also reported reluctance to take antidepressant medication because of the risk of triggering a manic

episode. Only about 75% of those who completed an assessment enrolled in a treatment course, and we do not know if the treatment sample was representative of all patients with BD who completed an assessment, or whether that created a further selection bias. Additional confounders included the potential effect of socioeconomic status on outcome, [48] as the BD group had a higher level of education, and remote location on the likelihood of being prescribed Lithium, because of the reduced availability of psychiatrists and the monitoring required for that treatment. However, iCBT has the potential to overcome inequalities in the provision of evidence based psychological care, as demonstrated by the good results from a related service for disadvantaged patients, which included more male subjects [49]. A further limitation is the probability that there were many patients in the total clinic sample with BD who were receiving treatment with other forms of mood stabilising medication, or no medication at all. The proportion who reported taking Lithium was only 0.57% of a large sample of people with clinically significant symptoms of depression, which as well as those taking other mood stabilisers, is likely to have included a proportion who were yet to be diagnosed with BD. However, the proportion with BD in the clinic sample is unlikely to have been greater than the figure of 7.4% of a national sample of depressed patients with as yet undiagnosed bipolar reported from Finland [5] and hence was unlikely to be large enough to affect the results. Moreover, the point of the study was to examine whether iCBT could be effective for depression in people with BD, and the specificity of the inclusion criteria was considered to be more important than the sensitivity.

Other limitations include the under-representation of males and the lack of detailed information about the participants, in particular comorbid conditions, such as substance use, or risk factors such as past trauma, which might have increased the relevance of psychological treatment, although the iCBT courses are largely agnostic to the causes of symptoms and instead focuses on recognising the presence of symptoms and willingness to change. A further consideration is the higher baseline symptoms of the BD group, which can translate to greater effect sizes in treatment [50]. However, the percentage changes in symptoms, which is a more conservative measure, were also significant. The significant limitations of this study in turn limit the generalisability of the findings, and the need for further research to confirm the findings. Moreover, there was no direct comparison of treatment outcomes because of the hugely unequal sample sizes and hence a direct comparison of the outcomes of BD and non-BD patients cannot be made.

With those limitations in mind, this study demonstrates the effectiveness of MindSpot courses for treating anxiety and depression in a sample of people with probable BDd, and confirms the effectiveness of iCBT delivered as part of routine care, as well as the potential of internet delivered mental health services to address the unmet need for treatment of depression in people with BD. The findings of this study also adds to the body of scientific evidence for the efficacy of CBT for the depressed phase of BD. Future research will need to examine the reliability of the diagnosis of BD, examine the outcome on BDd patients receiving treatment other than Lithium, include a matched comparison group for a better assessment of outcome, and a waitlist controlled trial for iCBT in patients with confirmed BDd. A question about the patient's understanding of the reason for taking medication has recently been added to MindSpot assessment questionnaire with a view to further evaluation. With those limitations, and the need for further research and evaluation in mind, services such as MindSpot, which are provided by trained therapists operating within an established clinical governance framework, should be seen as a treatment option alongside face to face mental health services, to ensure that people with bipolar disorder receive the full array of recommended treatments.

## Supporting information

**S1 Data. Self reported psychotropic medication and symptom scores for each patient.**
(XLSX)

## Acknowledgments

We would like to acknowledge the very high standard of care provided by Mindspot clinicians.

## Author Contributions

**Conceptualization:** Olav Nielssen, Lauren Staples, Blake Dear, Nickolai Titov.

**Data curation:** Lauren Staples, Eyal Karin, Rony Kayrouz.

**Formal analysis:** Lauren Staples.

**Supervision:** Blake Dear, Nickolai Titov.

**Writing – original draft:** Olav Nielssen.

**Writing – review & editing:** Lauren Staples, Eyal Karin, Rony Kayrouz, Blake Dear, Nickolai Titov.

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
