## [Decision Letter · Decision Letter 0]

19 Dec 2022

PDIG-D-22-00173

Effectiveness of internet delivered cognitive behaviour therapy provided as routine care for people in the depressed phase of bipolar disorder

PLOS Digital Health

Dear Dr. Nielssen,

Thank you for submitting your manuscript to PLOS Digital Health. After careful consideration, we feel that it has merit but does not fully meet PLOS Digital Health's publication criteria as it currently stands. Therefore, we invite you to submit a revised version of the manuscript that addresses the points raised during the review process.

Please submit your revised manuscript within 30 days Jan 18 2023 11:59PM. If you will need more time than this to complete your revisions, please reply to this message or contact the journal office at digitalhealth@plos.org. Please include the following items when submitting your revised manuscript:

We look forward to receiving your revised manuscript.

Kind regards,

Danilo Pani, Ph.D.

Academic Editor

PLOS Digital Health

Journal Requirements:

2. Please send a completed 'Competing Interests' statement, including any COIs declared by your co-authors. If you have no competing interests to declare, please state "The authors have declared that no competing interests exist". Otherwise please declare all competing interests beginning with the statement "I have read the journal's policy and the authors of this manuscript have the following competing interests:"s

3. Please amend your detailed Financial Disclosure statement. This is published with the article. It must therefore be completed in full sentences and contain the exact wording you wish to be published.

1. Please clarify all sources of funding (financial or material support) for your study. List the grants (with grant number) or organizations (with url) that supported your study, including funding received from your institution. 

2. State the initials, alongside each funding source, of each author to receive each grant.

3. State what role the funders took in the study. If the funders had no role in your study, please state: “The funders had no role in study design, data collection and analysis, decision to publish, or preparation of the manuscript.”

4. If any authors received a salary from any of your funders, please state which authors and which funders.

4. We have noticed that you have uploaded Supporting Information files, but you have not included a list of legends. Please add a full list of legends for your Supporting Information files after the references list. 

5. In the online submission form, you indicated that "All authors had access to all of the data. The datasets used and/or analysed during the current study available from the corresponding author on reasonable request". All PLOS journals now require all data underlying the findings described in their manuscript to be freely available to other researchers, either 1. In a public repository, 2. Within the manuscript itself, or 3. Uploaded as supplementary information.

Additional Editor Comments (if provided):

The manuscript reports an interesting piece of research, however, the authors are kindly asked to improve the presentation by considering the attached reviews. In particular, please take into account the comments provided by the third reviewer, both in the manuscript and in rebuttal letter.

Reviewers' comments:

Reviewer's Responses to Questions

**Comments to the Author**

1. Does this manuscript meet PLOS Digital Health’s publication criteria? Is the manuscript technically sound, and do the data support the conclusions? The manuscript must describe methodologically and ethically rigorous research with conclusions that are appropriately drawn based on the data presented.

Reviewer #1: Yes

Reviewer #2: Partly

Reviewer #3: Partly

2. Has the statistical analysis been performed appropriately and rigorously?

Reviewer #1: Yes

Reviewer #2: Yes

Reviewer #3: Yes

3. Have the authors made all data underlying the findings in their manuscript fully available (please refer to the Data Availability Statement at the start of the manuscript PDF file)?

Reviewer #1: No

Reviewer #2: Yes

Reviewer #3: Yes

4. Is the manuscript presented in an intelligible fashion and written in standard English?

Reviewer #1: Yes

Reviewer #2: Yes

Reviewer #3: Yes

5. Review Comments to the Author

Reviewer #1: This is a well considered piece of research and a well written paper. Small things first - There are a few typos in the results section of the abstract and "effictiveness" later in the text. Larger things - this is well done overall. My concern is that the conclusions that you state in the limitations section "with those limitations in mind.." (p17) are entirely fair and I think in order to accept this these should be the conclusions you quote in the Conclusions section of the abstract which are much more assertive and overly so in my opinion. Selection bias and the impact of SES as a confounder are well known issues in the analysis of technology in healthcare and are probably at work here. There is huge class imbalance between the intervention and control group that is also typical of such analyses and could plausibly limit the generalizability of the findings here - might be worth citing this and discussing that these issues are not unique to the analysis of this technology in the limitations section - https://www.tandfonline.com/doi/full/10.1080/13696998.2021.1890416. With all that in mind I think it is reasonable to ask that the conclusions stated in the abstract are more conservative.

Reviewer #2: Dear Authors:

Thank you for the opportunity to review this interesting and important paper. Bipolar depression has a devastating impact on affected individuals and is notoriously difficult to treat. Further understanding of the effectiveness of internet delivered cognitive behavior therapy as part of routine care will fill gaps in current knowledge to help providers and patients manage this debilitating disease. This paper is well-written, clear, and – despite significant limitations – does advance the science of the management of bipolar depression. In the spirit of constructing as strong a paper as possible to convey this important information, I would like to offer some specific edits to the paper. I ask the authors to consider these suggestions not as overly focused on small details; rather in the spirit of creating the strongest, most succinct prose possible to highlight the strength of this paper. 

Title: Please change the title to indicate that this study included only BDd patients who have been prescribed lithium. As the authors noted, this is a specific – but not sensitive – indicator of patients with BD. A large group of patients with BD have potentially been excluded from the study sample. It is important to clearly identify this in the title to assist with literature searches.

Abstract: No suggested revisions

Author summary: The conclusion “The study shows that Lithium treatment is almost exclusively reserved for bipolar disorder in Australia” is a generalization not supported by the evidence provided in this study. Please either reword or remove this sentence. 

Introduction: The sentence which begins “However, once the diagnosis has been established, the depressed phase of BD is then assumed …” is awkward. Please consider rewriting the sentence to improve readability.

Outcome measures: Please clarify why, if symptoms questionnaires were completed weekly, you chose to only use baseline and completion scores for analysis. 

Table 1: Please ensure accuracy between the text and the information provided in Table 1. I noted several discrepancies. For example, the text states “Those with confirmed BD were older (43.9 years, SD 13.3 vs 39.8 years, SD 13.8),” however the table shows Confirmed BD 43.8 (13.3); the text states “Those with confirmed BD were … less likely to be employed (46.3% versus 61.2%),” however the table indicates Confirmed BD 49.2%. These are only 2 examples of discrepancies I noted in review of the concordance between text and table.

Discussion: In the first sentence please include an indication that patients in the study were people who were prescribed lithium. 

Please indicate that AD is an abbreviation for antidepressant. 

The statement “The results of this study suggest a greater emphasis should be placed on psychological treatments for BDd…” is presumptive. Consider something that better reflects the findings of this study within the context of the limited available evidence such as, “The results of this study suggest a prominent role for psychological interventions in the treatment of BDd …”

In the middle of page 17 there is an abbreviation that I believe is typed incorrectly as “DBb”

Thank you for the opportunity to review this excellent study. With the revisions noted here, I believe this paper would make a valuable contribution to the growing body of scientific literature on BDd treatment.

Reviewer #3: Save for some observations which I believe can be sorted out, i think that the overall concept which informed the research is noteworthy and the result will help transform how we address the needs of theses group of patients. All other observations are as depicted in my critique. Thank you

6. PLOS authors have the option to publish the peer review history of their article (what does this mean?). If published, this will include your full peer review and any attached files.

**Do you want your identity to be public for this peer review?** For information about this choice, including consent withdrawal, please see our Privacy Policy.

Reviewer #1: Yes: Dr Trishan Panch

Reviewer #2: No

Reviewer #3: Yes: Paul Ede Agbo

---

## [Editor Report · Decision Letter 1]

9 Jan 2023

PDIG-D-22-00173R1

Effectiveness of internet delivered cognitive behaviour therapy provided as routine care for people in the depressed phase of bipolar disorder treated with Lithium

PLOS Digital Health

Dear Dr. Nielssen,

Thank you for submitting your manuscript to PLOS Digital Health. After careful consideration, we feel that it has merit but does not fully meet PLOS Digital Health's publication criteria as it currently stands. Therefore, we invite you to submit a revised version of the manuscript that addresses the points raised during the review process.

Please submit your revised manuscript within 30 days Feb 08 2023 11:59PM. If you will need more time than this to complete your revisions, please reply to this message or contact the journal office at digitalhealth@plos.org. Please include the following items when submitting your revised manuscript:

We look forward to receiving your revised manuscript.

Kind regards,

Danilo Pani, Ph.D.

Academic Editor

PLOS Digital Health

Journal Requirements:

Additional Editor Comments (if provided):

Dear Authors, the Reviewer 3 comments were (hopefully) attached. I'm attaching them again. Please consider also them.
---

## [Editor Report · Decision Letter 2]

13 Jan 2023

Effectiveness of internet delivered cognitive behaviour therapy provided as routine care for people in the depressed phase of bipolar disorder treated with Lithium

PDIG-D-22-00173R2

Dear Professor Nielssen,

We are pleased to inform you that your manuscript 'Effectiveness of internet delivered cognitive behaviour therapy provided as routine care for people in the depressed phase of bipolar disorder treated with Lithium' has been provisionally accepted for publication in PLOS Digital Health.

Best regards,

Danilo Pani, Ph.D.

Academic Editor

PLOS Digital Health

Overall, I think the authors provided satisfactory responses to the Reviewers' comments, though I would have expected to see some further explanation also in the manuscript.

As the authors recognize, "Plos Digital covers all health, not only mental health, and some more detailed description of our service might assist." This is the point: as this is not a vertical journal but a multidisciplinary one, some explanations that sound obvious to the authors do not necessarily are obvious for (a part of) the readers. There are people from very different research fields among the readers, and they have all the same dignity and right to understand the scientific papers. A few more words to be more clear cost very little to the authors and may be very useful for some readers.

About the rebuttal that "our service is one of the world leaders in the development and delivery of mental health treatments via the internet, and as stated in the response to reviewer 3, this paper is one of more than 100 publications by our team that are readily available in open access journals": in a double blind revision, this would have been impossible to assess and any paper from any author is relevant or not, clear or not, ... for its content and not for the importance of the authors. Maybe some comments in the manuscript would have been more helpful for some readers to better understand the same things that the Reviewer missed.

Finally, about the rebuttal on "Alleging that 3 out of 6 of the researchers have direct or indirect interest in the intervention making it “a difficult sell” is rather pejorative, when we have made it clear that the service is fully funded by the Australian government. None of the authors derive or seek to derive any pecuniary interest apart from our employment by the service." I agree that the Reviewer's wording was maybe rude, however I think the point was different: regardless who's paying their salary, and regardless the direct pecuniary interest, if some authors are "authors and developers of the courses offered at MindSpot" it is difficult they will present results which are against the quality of the intervention with such tools: it is not just a matter of money. I think the Reviewer's comment was in this sense only; however, this is clearly disclosed in the paper, even better in the revised version, and does not affect the results.

I hope these few lines will help the authors understanding some comments, my request, and the spirit of this multidisciplinary journal, which is expressly interested in this kind of research and in high-quality paper like this manuscrip.